# Stilbenoids and Flavonoids from *Cajanus cajan* (L.) Millsp. and Their *α*-Glucosidase Inhibitory Activities

**DOI:** 10.3390/molecules28093779

**Published:** 2023-04-27

**Authors:** Yaxian Zhao, Xinman Zhao, Mengjia Guo, Krishnapriya M. Varier, Babu Gajendran, Shaohuan Liu, Ling Tao, Xiangchun Shen, Nenling Zhang

**Affiliations:** 1The State Key Laboratory of Functions and Applications of Medicinal Plants, School of Pharmaceutical Sciences, Guizhou Medical University, Guiyang 550025, China; zhaoyaxian@stu.gmc.edu.cn (Y.Z.);; 2The High Efficacy Application of Natural Medicinal Resources Engineering Center of Guizhou Province, School of Pharmaceutical Sciences, Guizhou Medical University, Guiyang 550025, China; 3The Key Laboratory of Optimal Utilization of Natural Medicine Resources, School of Pharmaceutical Sciences, Guizhou Medical University, Guiyang 550025, China

**Keywords:** *Cajanus cajan*, stilbenoid, flavonoid, *α*-glucosidase inhibitory activity

## Abstract

Two new stilbenoids, cajanstilbenoid C (**1**) and cajanstilbenoid D (**2**), together with eight other known stilbenoids (**3**-**10**) and seventeen known flavonoids (**11**-**27**), were isolated from the petroleum ether and ethyl acetate portions of the 95% ethanol extract of leaves of *Cajanus cajan* (L.) Millsp. The planar structures of the new compounds were elucidated by NMR and high-resolution mass spectrometry, and their absolute configurations were determined by comparison of their experimental and calculated electronic circular dichroism (ECD) values. All the compounds were assayed for their inhibitory activities against yeast *α*-glucosidase. The results demonstrated that compounds **3**, **8**-**9**, **11**, **13**, **19**-**21,** and **24**-**26** had strong inhibitory activities against *α*-glucosidase, with compound **11** (IC_50_ = 0.87 ± 0.05 μM) exhibiting the strongest activity. The structure–activity relationships were preliminarily summarized. Moreover, enzyme kinetics showed that compound **8** was a noncompetitive inhibitor, compounds **11**, **24**-**26** were anticompetitive, and compounds **9** and **13** were mixed-competitive.

## 1. Introduction

Diabetes mellitus (DM) is a metabolic disorder induced by multiple causes and characterized by persistent hyperglycemia. With improved living standards and altered lifestyles, the number of diabetes patients is on the rise and is expected to reach 300 million by 2025, which will rank diabetes as the third most common disease that seriously endangers human health and causes a huge economic burden, right after cardiovascular diseases and malignant cancer [1]. Diabetes is categorized into types I and II, and the number of patients suffering from type II accounts for more than 90% [2]. The available oral drugs in clinical use for the treatment of diabetes include biguanides, sulfonylureas, insulin sensitizers, glinide insulin secretagogue, *α*-glucosidase inhibitors, dipeptidyl peptidase-IV (DPP-4) inhibitors, and sodium-glucose cotransporter 2 (SGLT-2) inhibitors [3]. *α*-Glucosidase inhibitors exert their hypoglycemic effect by competitively inhibiting *α*-glucosidase located in the small intestine to slow down the decomposition of starch into glucose. At present, α-glucosidase inhibitors mainly include acarbose, voglibose, and miglitol [4]. However, the preparation process of those oral hypoglycemic agents is complicated and the cost for long-term use is high, and their long-term use often causes abdominal discomforts [5]. Therefore, it is urgent to find new *α*-glucosidase inhibitors that might eliminate these problems.

*Cajanus cajan* (L.) Millsp, an erect shrub or subshrub, also known as pigeon pea, is the sixth-largest edible bean in the world. *C. cajan* is mainly distributed in South and Southeast Asia, in countries such as India, Myanmar [6], and China. In China, it is used as a traditional Chinese medicine to treat wounds, malaria, coughs, and abdominal pain [7]. Chemical studies show that it mainly contains stilbenes and flavonoids, as well as triterpenoids, steroids, lignans, and alkaloids, and its crude extract or compounds isolated from it are reported to have antioxidant, anticancer, antibacterial, anti-inflammatory, antimalarial, lipid-regulating, cognitive-enhancing, antiosteoporosis, and antidiabetic properties [8,9,10].

Our interest in finding *α*-glucosidase inhibitors led to the purification of two new stilbenoids, and twenty-five known compounds from the leaves of *C. cajan* (Figure 1). All the compounds were tested for their *α*-glucosidase inhibitory activity. Compound **11** (IC_50_ = 0.87 ± 0.05 μM) had the strongest activity, and compounds **13**, **20**, **25**-**26** also showed remarkable activities, with IC_50_ values at 2.3 ± 0.50, 4.5 ± 0.49, 3.17 ± 0.55 and 2.35 ± 0.54, respectively. Furthermore, the structure–activity relationships of these compounds were preliminarily summarized, and the enzyme kinetics of active compounds were studied.

## 2. Results and Discussion 

Based on our previous results from the same plant collected from Guizhou [10], flavonoids might be promising α-glucosidase inhibitors, and they are mainly present in petroleum ether and ethyl acetate parts extracted from 95% ethanol crude extract; thus, herein, the dried leaves of *C. cajan* (30 kg) were extracted with 95% ethanol. The extraction solvent was then evaporated with a rotatory evaporator to give a crude extract; the crude extract was dissolved in water and successively partitioned with petroleum ether and ethyl acetate to afford petroleum ether part (835.6 g) and ethyl acetate part (456.4 g). Petroleum ether and ethyl acetate extracts being subjected to purification processes yielded ten stilbenoids (**1**-**10**) and seventeen flavonoids (**11**-**27**). The structures of new compounds (**1**-**2**) were characterized by HR-MS,1D-NMR, 2D-NMR, and ECD, while the structures of known compounds (**3**-**27**) were determined mainly through comparison of 1D-NMR data with those reported in the literature. 

In order to obtain promising α-glucosidase inhibitors, all the purified compounds were screened for their α-glucosidase inhibitory activity, and the ones that showed robust activity were further elucidated for their inhibition type by Lineweaver–Burk plots.

### 2.1. Structural Elucidation 

Compound **1** was a yellow powder. From its quasi-molecular ion peak at m/z 377.1354 (calcd. for m/z 377.1359, [C_21_H_22_O_5_Na]^+^) on positive ion HR-ESI-MS (Appendix A), its molecular formula was indicated as C_21_H_22_O_5_; thus, an unsaturation degree of 11 was calculated. By comparing both its ^1^H-NMR and ^13^C-NMR data (Table 1 and Appendix A) with those of other known stilbenes, we isolated (**3**-**10**); we suspected it was also a derivative of stilbene. In the ^1^H-NMR data, the signal at *δ*_H_ 6.45 (1H, s) and peaks at *δ*_H_ 7.43 (2H, d, *J* = 6.9 H_Z_), 7.34 (2H, t, *J* = 7.6 Hz), and 7.24 (1H, t, *J* = 7.4 Hz) suggested the existence of two benzene rings. Two methyl groups were indicated according to the peaks of *δ*_H_ 0.68 (s, 3H), 0.66 (s, 3H) on ^1^H-NMR, and *δ*_C_ 30.1, 25.0 on ^13^C-NMR, which, together with signals [*δ*_H_ 3.21 (1H, d, *J* = 16.0 Hz), 2.58 (1H, dd, *J* = 16.0, 8.4 Hz), 2.46 (1H, t, *J* = 8.1 Hz), *δ*_C_ 73.1, 51.3, 21.2], implied the presence of an oxidized isopentyl unit. Peaks at *δ*_H_ 3.85 (3H, s) and *δ*_C_ 56.4 suggested the presence of a methoxy group. The signal at *δ*_C_ 168.4 indicated the presence of a carbonyl group. In addition, the NMR data were in great similarity with those of Carexane O [11,12], suggesting that the skeleton of **1** was the same as Carexane O, which was also supposed to be biogenetically derived from stilbene. The differences between **1** and Carexane O were that instead of the presence of two hydroxy groups on the benzene ring, one of the hydroxyls changed into a methoxy group, and there was also an additional carbonyl group. Moreover, the unsaturation number of **1** is **11**, and Carexane O is 10, which implies that instead of a free hydroxy group on the C2 between the two benzene rings, it might be cyclized into an ester ring with its adjacent hydroxy group, which was confirmed by the molecular weight of 354 (deduced from its quasi-molecular ion at 377.1354 [M+Na]^+^). To further evidence this cyclization, HMBCs of *δ*_H_ 5.22/*δ*_C_ 162.1, 145.5 were found, and other key HMBCs are also shown in Figure 2. Thus, the planar structure of compound **1** was established. The configuration of H-7 and H-8 was deduced to be trans according to their large H-H coupling constant (*J* = 10.5 Hz), while the configuration of H-8 and H-2” was elucidated to be cis by the correlation [2.30 (dd, *J* = 10.5, 7.7 Hz, H-8)/2.46 (t, *J* = 8.6 Hz, H-2”)] found on the NOESY spectrum (Appendix A). The experimental ECD spectrum of compound **1** agreed with its calculated data of the (7*R*, 8*R*, 2”*S*) configuration (Figure 3a); therefore, the absolute configuration of compound **1** (structure shown in Figure 1) was elucidated as (7*R*, 8*R*, 2”*S*). Thus, compound **1** was determined to be new, and given the name of cajanstilbenoid C.

Compound **2** was yellow amorphous powder; from its quasi-molecular ion peak at m/z 377.1354 (calcd. for m/z 377.1359, [C_21_H_22_O_5_Na]^+^) on positive HR-ESI-MS (Appendix A), its molecular formula was indicated as C_21_H_22_O_5_, and an unsaturation degree of 11 was calculated. In the ^1^H-NMR data (Table 1 and Appendix A), signals from the low-field region suggested the presence of penta- [*δ*_H_ 6.90 (1H, s)] and monosubstituted [*δ*_H_ 7.50 (2H, d, *J* = 7.3 Hz), 7.38 (2H, t, *J* = 7.6 Hz), 7.29 (1H, s)] benzene rings. The peaks of *δ*_H_ 7.08 (1H, d, *J* = 16.3 Hz) and *δ*_H_ 7.29 (1H, d, *J* = 15.1 Hz), with their large coupling constants, suggested that they belonged to two trans-protons on a double bond. The above information led to the elucidation that compound **2** is also a stilbene derivative. In addition, resonances at *δ*_H_ 3.88 (3H, s) and *δ*_C_ 56.1 signified a methoxy group. The signal of *δ*_C_ 169.4 implied the presence of a carbonyl. Resonances of *δ*_H_ 2.41 (1H, dd, *J* = 17.4, 7.6 Hz), 2.79 (1H, dd, *J* = 17.4, 5.5 Hz), 3.65-3.62 (1H, m), 1.25 (3H, s), and 1.14 (3H, s) on the ^1^H-NMR, with their corresponding signals at *δ*_C_ 26.8, 67.9, 77.7, 25.9, and 20.6 on the ^13^C-NMR (Table 1), suggested the presence of a hydroxylated isopentyl moiety, which was cyclized into a six-membered ring with its ortho hydroxy group. The NMR data of **2** were found to resemble those of Chiricaine B [13], except that for **2**, a hydroxy group changed into a methoxy group, and there was an additional signal of a carbonyl. The location of the methoxy group was determined by the following HMBC correlations: *δ*_H_ 3.88/*δ*_C_ 157.9; *δ*_H_ 2.41 (1H, dd, *J* = 17.4, 7.6 Hz), 2.79 (1H, dd, *J* = 17.4, 5.5 Hz)/*δ*_C_ 157.9. The location of the carbonyl was settled by the HMBC correlation of *δ*_H_ 6.90 (1H, s) with *δ*_C_ 109.4 (key HMBC correlations are shown in Figure 2). The experimental ECD spectrum of compound **2** agreed with its calculated data of 2”*R* configuration (Figure 3b), so the absolute configuration of compound **2** (structure shown in Figure 1) was elucidated as 2”*R*. Thus, compound **2** was determined to be new, and given the name cajanstilbenoid D.

The known compounds (**3**-**27**, Figure 1) were identified as chiricanine A (**3**) [14], chiricanine B (**4**) [13], cajaninstilbene acid (**5**) [15], pinosylvin monomethyl ether (**6**) [16], cajanotone (**7**) [17], 3-methoxy-5-hydroxy-2-(3-methyl-2-butenyl) bibenzyl (**8**) [18], longistylin A (**9**) [19], (E)-methyl-2-hydroxy-4-methoxy-6-styrylbenzoate (**10**) [20], 8-prenylquercretin (**11**) [21], 6-C-(3,3-dimethylallyl) chrysin (**12**) [22], 5-hydroxy-7-methoxy-8-prenylflavone (**13**) [23], 3, 5-dihydroxy-7-methoxy- 8-isopentenyl-dihydroflavone (**14**) [24], pinostrobin (**15**) [25], sakuranetin (**16**) [26], diosmetin (**17**) [27], 7-methoxy-2′, 4′-dihydroxy isoflavone (**18**) [28], tamarixetin (**19**) [29], neophellamuretin (**20**) [30], 3, 4′-O-dimethylquercetin (**21**) [31], quercetin-3-*O*-α-L-rhamnopyranoside (**22**) [32], apigenin-8-C-α-L-arabinopyranoside (**23**) [33], 7, 3′, 4′-trihydroxy isoflavone (**24**) [34], 7, 3′, 4′-trihydroxy flavone (**25**) [35], 8-prenylnaringenin C (**26**) [36], and alpinum isoflavone (**27**) [37] by comparing their spectroscopic data (Appendix A) with those from previous reports.

### 2.2. α-Glucosidase Inhibitory Activity of Compounds ***1-27***

All the isolated compounds (**1**-**27**) were evaluated for their *α*-glucosidase inhibitory activity; among them, compounds **3**, **8**-**9**, **11**, **13**, **19**-**21,** and **24**-**26** efficaciously inhibited *α*-glucosidase, in which compound **11** (IC_50_ = 0.87 ± 0.05 μM) displayed the strongest inhibitory activity towards *α*-glucosidase, and the IC_50_ value of the positive control (acarbose) was 352.87 ± 0.82 (Table 2).

### 2.3. Inhibitory Kinetics of Compounds ***8-9***, ***11***, ***13***, ***24***-***26*** against α-Glucosidase

The inhibitory kinetics of seven compounds (**8**-**9**, **11**, **13**, **24**-**26**) that have strong *α*-glucosidase inhibitory activities were studied. The Lineweaver–Burk plot was used to determine their inhibition types against *α*-glucosidase. As shown in Figure 4, all the lines of compounds **11**, **24**, **25,** and **26** are almost parallel. As the concentrations of compounds **11**, **24**, **25,** and **26** decreased, V_max_ (maximum reaction rate) and K_m_ (Michaelis constant) decreased, but K_m_/V_max_ remained unchanged; in this case, the way compounds **11**, **24**, **25,** and **26** mediated *α*-glucosidase inhibition was anticompetitive, which means that the compounds do not directly bind to the free enzyme, but only bind to the enzyme–substrate complex, thereby interrupting the enzymatic reaction. As for compound **8**, the slope of the lines became greater with the increase in the concentration, but all straight lines intersected with the x-axis almost at one point (while V_max_ decreased, K_m_ was unchanged), showing that *α*-glucosidase inhibition mediated by compound **8** is noncompetitive, signifying that it can bind to either the free *α*-glucosidase or the enzyme–substrate complex to interfere the enzyme reaction without directly blocking the binding of the substrate to the enzyme. For compounds **9** and **13**, the slope of the lines also became greater when their concentrations increased, but all the lines intersected in the third quadrant (K_m_ decreased with V_max)_, indicating that compounds **9** and **13** mediated *α*-glucosidase inhibition in a mixed way, in which case, inhibitors can bind to either the enzyme or enzyme–substrate complex.

### 2.4. Structure–Activity Relationship Analysis 

In this study, two classes of compound, stilbenoids (**1**-**10**) and flavonoids (**11**-**27**), have been isolated and identified. By comparing their IC_50_ values (Table 2), the inhibitory effects of flavonoids against *α*-glucosidase were stronger than those of the stilbenoids, which might be due to their different skeletons.

For stilbenoids, the activity of the ones that have hydroxy and methoxy substituents at C-3 and C-5 is relatively high, and the activity of the one possessing a methoxy group at C-5 is stronger than the one with a hydroxyl group at this position. In addition, the isopentenyl substituent favors activity, and when ring A contains a carboxyl or ester group, the activity decreases.

For flavonoids, their type (flavones or isoflavones), glycosylation, and substituents (positions of hydroxy, methoxy groups, and isopentenyl moiety) all affect their inhibitory effects against *α*-glucosidase. Basically, the activity of flavones was stronger than that of isoflavones, and when the hydroxyl groups were substituted at C-5 and C-7, and the isopentenyl group was substituted at C-8, the inhibition effect was enhanced, whereas the activity was reduced when the hydroxyl group was glycosylated. The activity of the B ring with the hydroxyl group on C-3′ and C-4′was stronger than that of no substituent on the B ring.

## 3. Materials and Methods

### 3.1. General Experimental Procedures

NMR data were obtained by Bruker Avance Neo–400 MHz NMR spectrometer (Bruker, Germany); a VG-Autospec-3000 mass spectrometer (Beckman Coulter, Inc. America) was adopted to acquire HR-ESIMS spectra. A Fourier transform infrared spectrometer FTIR-650 from Tianjin Gangdong Technology Development Co., Ltd. (Tianjin, China) was employed to perform infrared spectra. UV spectra were documented by UV/Vis spectrophotometer UV-2700 (Shimadzu Instrument Suzhou Co., Ltd., Suzhou, China). TLC plates of Silica gel GF254 were purchased from Yantai Jiangyou Silicon Development Company (Yantai, China), and spots were observed by being exposed under UV light or heated after being sprayed with H_2_SO_4_ dissolved in EtOH (5% *v*/*v*). Purification by HPLC was carried out with LC-20AR pumps and an SPD-M20A UV detector (Shimadzu, Kyoto, Japan). With a J-810 CD spectrometer from JASCO, Ltd. (Tokyo, Japan), ECD spectra were measured. The optical rotation was detected under the polarimeter-Autopol VI (Yunnan Gaosheng Import & Export Co., Ltd., Yunnan, China). A BioTek ELX800 microplate reader (USA) was used to measure absorbance.

Chemicals: *α*-Glucosidase enzyme (Saccharomyces cerevisiae) was acquired from Macklin (Shanghai, China); the substrate (*p*-Nitrophenyl *α*-D-glucopyranoside, p-NPG) and acarbose (positive drug) were acquired from Aladdin (Shanghai, China) and Sigma Aldrich Co., St. Louis, MO, USA, respectively.; methanol, 95% ethanol, and methylene chloride were purchased from Chongqing Chuandong Chemical Group Co., Ltd.; petroleum ether, ethyl acetate, chloroform, and acetone were from Sinopharm Chemical Reagent Co., Ltd.; methanol for HPLC purification was acquired from TEDIA.

### 3.2. Plant Material

The plant materials were purchased on 29 September 2020 from Taifu Agriculture and Forestry Technology Company (Honghe County, Yunnan Province, China), and identified by associate professor Shaohuan Liu to be leaves of *C. cajan* (L.) Millsp. A voucher specimen (20200929) was preserved at the Herbarium of School of Pharmaceutical Sciences, Guizhou Medical University. 

### 3.3. Extraction and Isolation 

The leaves of *C. cajan* (30 kg) were extracted by reflux with 95% ethanol 3 times, for 2 h each time, and the combined ethanol was concentrated to obtain crude extract. After being suspended with water, the crude extract was partitioned with petroleum ether and ethyl acetate successively to obtain petroleum ether part (835.6 g) and ethyl acetate part (456.4 g).

Ethyl acetate part (456.4 g) was subjected to silica gel (100–200 mesh) column, eluted by petroleum ether/ethyl acetate (95:5 → 0:100); the eluate was detected by thin-layer chromatography to afford Fr. E1~Fr. E10. Then, Fr. E1 underwent silica gel column chromatography (petroleum ether: ethyl acetate 9:1), affording Fr. E1-1~Fr. E1-5, and Fr. E1-5 was further subjected to silica gel column chromatography (petroleum ether: ethyl acetate 100:1) to obtain Fr. E1-5-1~Fr. E1-5-11. Subfractions (Fr. E1-5-7-1 to Fr. E1-5-7-10) of Fr. E1-5-7 were obtained by silica gel column chromatography (petroleum ether: chloroform 9:1), in which Fr. E1-5-7-6 was recrystallized in methanol to obtain compound **3** (16.28 mg). Compound **5** (12.30 mg, t_R_ = 25 min) was obtained by HPLC (70% methanol) from Fr. E1-5-7-10, and Fr. E1-5-10-1~Fr. E1-5-10-11 were obtained from Fr. E1-5-10 by silica gel column chromatography (petroleum ether: acetone 95:5), and further purified by HPLC (80% methanol) to yield compound **4** (13.05 mg, t_R_ = 28 min). Fr. E1-5-8 was isolated by Sephadex LH-20 column chromatography (methanol), silica gel column chromatography (petroleum ether: chloroform 85:15), and HPLC (90% and 80% methanol) to afford compounds **6** (15.30 mg, t_R_ = 32 min) and **15** (8.79 mg, t_R_ = 40 min). Fr. E3 was subjected to silica gel column chromatography (petroleum ether: ethyl acetate 95:5) to obtain Fr. E3-1~Fr. E3-9, then Fr. E3-2 was subjected to silica gel column chromatography (petroleum ether: chloroform 95:5) and HPLC (75% methanol) to afford compound **13** (9.05 mg, t_R_ = 42 min). Fr. E3-6 was subjected to a Sephadex LH-20 column (methanol) to obtain Fr. E3-6-1~Fr. E3-6-4, then Fr. E3-6-4 was subjected to silica gel column (petroleum ether: ethyl acetate 95:5) and then a Sephadex LH-20 column (methanol) to obtain compound **9** (12.35 mg). Fr. E3-7 underwent silica gel column chromatography (petroleum ether: ethyl acetate 95:5) and then HPLC (70% methanol) to afford compound **10** (10.05 mg, t_R_ = 45 min). Fr. E4 was isolated by MCI column chromatography eluted by methanol–water gradient (50:50→100:0) to obtain Fr. E4-1~Fr. E4-7; Fr. E4-3 was isolated by Sephadex LH-20 column chromatography (methanol) to obtain Fr. E4-3-1~Fr. E4-3-8, in which compounds **8** (9.80 mg, t_R_ = 28 min) and **16** (8.50 mg, t_R_ = 35 min) were obtained by further silica gel column chromatography (petroleum ether: acetone 8:2) and HPLC (80% methanol). Fr. E4-2 underwent Sephadex LH-20 column chromatography (methanol) and HPLC (80% methanol) to obtain **14** (12.30 mg, t_R_ = 28 min). Compounds **7** (8.50 mg, t_R_ = 33 min) and **12** (10.05 mg, t_R_ = 41 min) were produced from Fr. E4-7 by Sephadex LH-20 column chromatography (methanol), silica gel column chromatography (petroleum ether: acetone 95:5), and HPLC (70% methanol). Fr. E6 was eluted by methanol–water gradient (30:70→100:0) on an MCI column to afford Fr. E6-1~Fr. E6-9. Then, Fr. E6-2 was separated by silica gel column chromatography (petroleum ether: ethyl acetate 8:2) and HPLC (50% methanol) to afford compound **11** (20.08 mg, t_R_ = 52 min). Fr. E6-4 was chromatographed over a silica gel column (chloroform: acetone 150:1) and Sephadex LH-20 column (methanol) to obtain compound **22** (8.90 mg). Fr. E6-8 was chromatographed on a silica gel column (petroleum ether: acetone 8:2) to yield Fr. E6-8-1~Fr. E6-8-15; Fr. E6-8-3 was recrystallized by methanol to furnish compound **17** (13.20 mg). Compound **18** (10.02 mg) was obtained by Sephadex LH-20 column chromatography (methanol) from Fr. E6-8-8. Compounds **1** (9.03 mg, t_R_ = 22min), **2** (10.20 mg, t_R_ = 34 min), **19** (9.80 mg, t_R_ = 40 min), **20** (9.50 mg, t_R_ = 44 min), and **21** (8.70 mg, t_R_ = 56 min) were isolated from Fr. E6-8-10 by repeated Sephadex LH-20 column (methanol) chromatography and HPLC (50–65% methanol). Fr. E7 was submitted to MCI column chromatography eluted with methanol–water gradient (20:80→100:0) to obtain Fr. E7-1~Fr. E7-7. Then, Fr. E7-3 underwent silica gel column chromatography (chloroform: ethyl acetate 9:1), Sephadex LH-20 column chromatography (methanol), and HPLC (55% methanol) to give compound **23** (12.80 mg, t_R_ = 62 min). 

The part of petroleum ether (835.6 g) was eluted with a gradient mixture of petroleum ether: ethyl acetate (100:0→30:70) on a silica gel column (100–200 mesh) to furnish Fr. M1~Fr. M8. Then, Fr. M4 was chromatographed over silica gel column (petroleum ether: dichloromethane 100:1) to obtain Fr. M4-1~Fr. M4-8; Fr. M4-3 was subjected to Sephadex LH-20 column chromatography (methanol) and HPLC (85% methanol) to obtain compound **24** (13.80 mg, t_R_ = 22 min), and compound **25** (16.40 mg, t_R_ = 28 min) was obtained by Fr. M4-6 over silica gel column chromatography (petroleum ether: ethyl acetate 9:1) and HPLC (80% methanol). Fr. M6 underwent silica gel column chromatography (petroleum ether: ethyl acetate 85: 25) to give Fr. M6-1~Fr. M6-10, in which Fr. M6-3 underwent repeated Sephadex LH-20 column chromatography (methanol), HPLC (80% methanol), and recrystallization in acetone to furnish compounds **26** (9.80 mg, t_R_ = 30 min) and **27** (12.50 mg, t_R_ =36 min).

### 3.4. Spectral and Physical Data of Compounds ***1*** and ***2***

Cajanstilbenoid C (**1**): Yellow amorphous powder. [*α*]$\begin{array}{l} 20\\ D \end{array}$ = + 25.455 (c = 0.0022, methanol). UV (methanol): λmax (log ɛ) 299 (0.85), 257 (0.87), 229 (2.03). IR: ν_max_ 3450, 3375, 3175, 2850, 1700, 1625, 1450, 1300, 1187, 1120, 1007, 831, 725, 684 cm ^−1^. HR-ESI-MS *m/z* 377.1354 [M+Na]^+^ (calcd. for *m/z* 377.1359, [C_21_H_22_O_5_Na]^+^); ^1^H and ^13^C-NMR data, see Table 1.

Cajanstilbenoid D (**2**): Yellow amorphous powder. [*α*]$\begin{array}{l} 20\\ D \end{array}$ = + 4.211 (c = 0.0019, methanol). UV (methanol): λmax (log ɛ) 306 (0.72), 236 (0.26). IR: *ν*max 3305, 3200, 2871, 2306, 1647, 1565, 1317, 1089, 1024, 935, 798, 733, 680 cm^−1^. HR-ESI-MS m/z 377.1354 [M+Na]^+^ (calcd. for *m/z* 377.1359, [C_21_H_22_O_5_Na]^+^); ^1^H and ^13^C-NMR data, see Table 1.

### 3.5. Assay for α-Glucosidase Inhibitory Activity of Compounds ***1***-***27***

The compounds **1**-**27** were evaluated for their inhibitory activities against *α*-glucosidase by employing the method of Lei [10]. This experiment was carried out in 96-well plates with reaction system of 230 μL. Firstly, 95 μL of PBS buffer solution (0.1 mol/L pH 6.8) was put into each well, then 5 μL of different concentrations (0.0625, 0.125, 0.25, 0.5, 1.0, 2.0 mmol/L) of acarbose solution or compounds **1**-**27** were added, then 30 μL of 0.6 U/mL α-glucosidase was pipetted and mixed by light shake. The 96-well plates were incubated in a constant-temperature incubator for 20 min at 37 °C, then 20 μL of 2.658 mmol/L *p*-PNG was added, and then incubation continued for another 28 min. The termination of the reaction was conducted by adding 80 μL of 0.2 mol/L Na_2_CO_3_ to each well. The absorbance was recorded for each well at 405 nm by a microplate reader, and the experiment was carried out three times in parallel. The measured absorbance values were processed by Graphpad Prism software 9.5 to calculate the IC_50_ values of the compounds. 

### 3.6. Kinetics of Compounds ***8***-***9***, ***11***, ***13***, ***24***-***26*** Inhibiting α-Glucosidase 

Concentrations of active compounds **8**-**9**, **11**, **13**, **24**-**26** (0, 5.44, 10.87, 21.74 μM) and the substrate *p*-NPG (400, 800, 1200, 1600, 2000 mg/mL) were prepared, and an experimental protocol is described in detail in the literature [10]. The data were acquired using the Lineweaver–Burk equation of enzyme kinetics.

## 4. Conclusions

In this study, two new compounds, named cajanstilbenoid C (**1**) and cajanstilbenoid D (**2**), together with eight other known stilbenoids (**3**-**10**) and seventeen known flavonoids (**11**-**27**), were isolated. *α*-Glucosidase inhibition assay of all the compounds showed that **3**, **8**-**9**, **11**, **13**, **19**-**21**, and **24**-**26** demonstrated robust inhibitory activity. By comparing their IC_50_ values, the inhibitory effects of flavonoids against α-glucosidase were stronger than those of the stilbenoids, which might be due to their different skeletons. For stilbenoids, the activity of the ones that have hydroxy and methoxy substituents at C-3 and C-5 is relatively high, and the activity of the one possessing a methoxy group at C-5 is stronger than the one with a hydroxyl group at this position. In addition, the isopentenyl substituent favors activity, and when ring A contains a carboxyl or ester group, the activity decreases. For flavonoids, their type (flavones or isoflavones), glycosylation, and substituents (positions of hydroxy, methoxy groups, and isopentenyl moiety) all affect their inhibitory effects against α-glucosidase. Basically, the activity of flavones was stronger than that of isoflavones, and when the hydroxyl groups were substituted at C-5 and C-7, and the isopentenyl group was substituted at C-8, the inhibition effect was enhanced, whereas the activity was reduced when the hydroxyl group was glycosylated. The activity of the B ring with the hydroxyl group on C-3′ and C-4′ was stronger than that of no substituent on the B ring. 

The inhibition types of compounds **8**–**9**, **11**, **13**, and **24**-**26** with strong activities were explored through enzymatic kinetics. Compound **8** was noncompetitive inhibitor, whereas compounds **11**, **24**-**26** were anticompetitive and compounds **9** and **13** were mixed ones.

This study enriched the number of compounds isolated from the leaves of *Cajanus cajan* (L.) Millsp. and revealed some promising *α*-glucosidase inhibitors, which could be helpful for better utilization of this plant; yet, more research is needed to demonstrate if the *α*-glucosidase inhibitors obtained herein could work in cell or animal models to treat diabetes. 

## Figures and Tables

**Figure 1 molecules-28-03779-f001:**
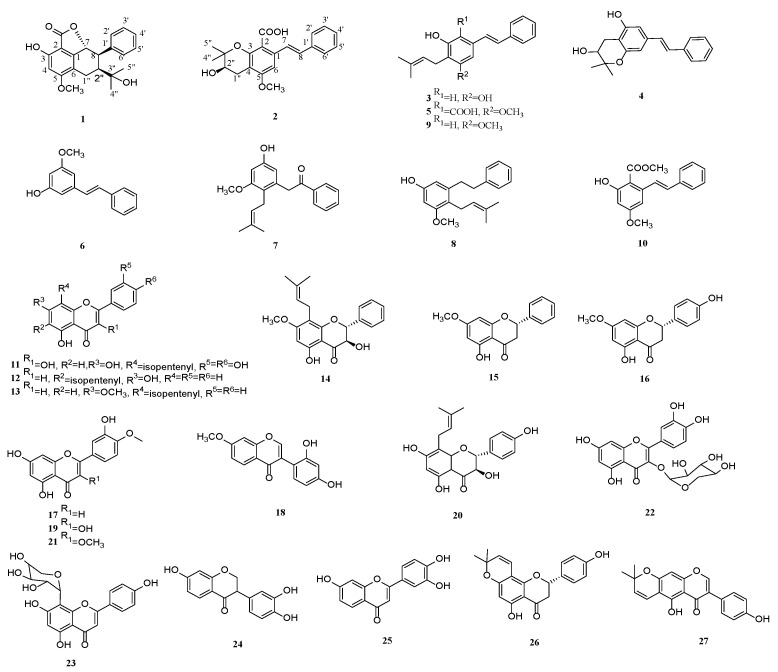
The structures of compounds **1**-**27**.

**Figure 2 molecules-28-03779-f002:**
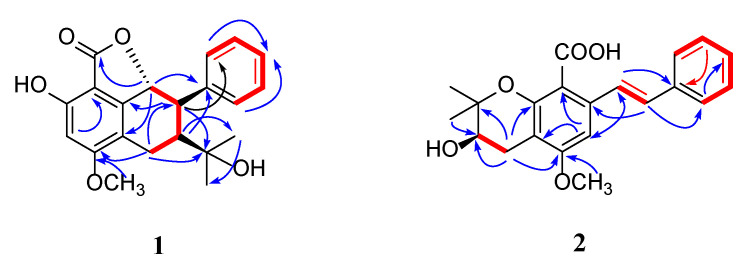
The key HMBC (
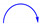
) and ^1^H—^1^H COSY (
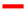
) correlations of compounds **1** and **2**.

**Figure 3 molecules-28-03779-f003:**
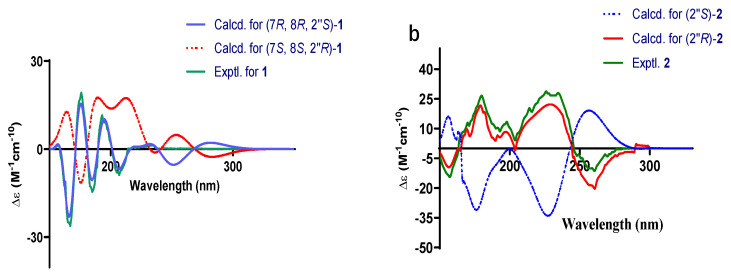
ECD spectra of compounds **1** (**a**) and **2** (**b**).

**Figure 4 molecules-28-03779-f004:**
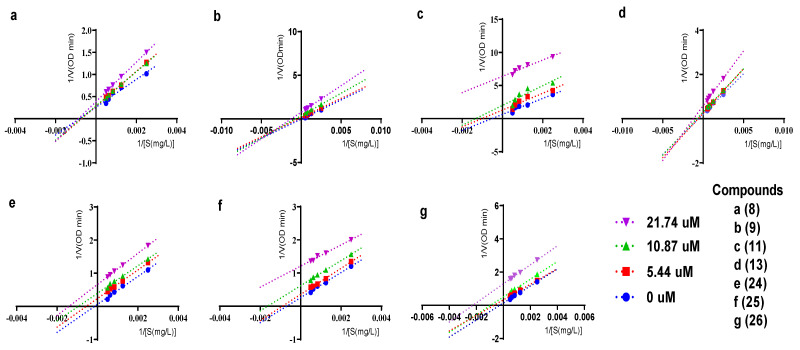
Lineweaver−Burk plots for the inhibition of α-glucosidase by compounds **8** (**a**)−**9** (**b**), **11** (**c**), **13** (**d**), and **24**−**26** (**e**−**g**).

**Table 1 molecules-28-03779-t001:** ^1^H-NMR (600 MHz) and ^13^C-NMR (150 MHz) data of compound **1**-**2** (*δ* in ppm, *J* in Hz, dissolved in (D6) DMSO).

Position	1		2	
	δ_H_	δ_C_	δ_H_	δ_C_
1	-	152.0	-	132.9
2	-	101.8	-	109.4
3	-	156.9	-	150.4
4	6.45 (s, 1H)	99.4	-	119.2
5	-	162.1	-	157.9
6	-	112.8	6.90 (s, 1H)	98.3
7	5.22 (d, *J* = 10.5 Hz, 1H)	82.0	7.29 (d, 1H, *J* = 15.1 Hz, overlapped)	128.4
8	2.30 (dd, *J* = 10.5, 7.7 Hz, 1H)	49.2	7.08 (d, *J* = 16.3 Hz, 1H)	125.9
1′	-	145.5	-	137.3
2′, 6′	7.43 (d, *J* = 6.9 Hz, 2H)	129.3	7.50 (d, *J* = 7.3 Hz, 2H)	126.8
3′, 5′	7.34 (t, *J* = 7.6 Hz, 2H)	128.7	7.38 (t, *J* = 7.6 Hz, 2H)	129.3
4′	7.24 (t, *J* = 7.4 Hz, 1H)	126.7	7.29 (s, 1H, overlapped)	130.3
1”	2.58 (dd, *J* = 16.1, 8.3 Hz, 1H);	21.2	2.41 (dd, *J* =17.4, 7.6 Hz, 1H);	26.8
	3.21 (d, *J* = 16.1 Hz, 1H)		2.79 (dd, *J* = 17.4, 5.5 Hz, 1H)	
2”	2.46 (t, *J* = 8.6 Hz, 1H)	51.3	3.62–3.65 (m, 1H)	67.9
3”	-	73.1	-	77.7
4”	0.66 (s, 3H)	30.1	1.25 (s, 3H)	25.9
5”	0.68 (s, 3H)	25.0	1.13 (s, 3H)	20.6
-OCH_3_	3.85 (s, 3H)	56.4	3.88 (s, 3H)	56.1
-COO-	-	168.4	-	169.4

**Table 2 molecules-28-03779-t002:** Inhibition activities of compounds **1**–**27** on α-Glucosidase (IC_50_ ± SD μM).

Compound	IC_50_ (μM)	Compound	IC_50_ (μM)
1	>2000	15	>2000
2	>2000	16	>2000
3	30.3 ± 2.33	17	>2000
4	>2000	18	>2000
5	>2000	19	8.7 ± 0.08
6	>2000	20	4.5 ± 0.49
7	>2000	21	19.4 ± 1.02
8	7.0 ± 0.45	22	>2000
9	12.9 ± 1.31	23	>2000
10	>2000	24	6.5 ± 0.42
11	0.87 ± 0.05	25	3.17 ± 0.55
12	>2000	26	2.35 ± 0.54
13	2.3 ± 0.51	27	>2000
14	>2000	Acarbose	352.87 ± 0.82

## Data Availability

Data of the compounds are available in Appendix A.

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
