# Peer review of "Stilbenoids and Flavonoids from Cajanus cajan (L.) Millsp. and Their α-Glucosidase Inhibitory Activities"

_molecules, 2023, doi:10.3390/molecules28093779_

Round 1

Reviewer 1 Report

The need for the research is well justified and the approach, results, and potential impact of the findings are clearly articulated. However, the authors should provide more details in the results section about the extraction process - why these solvents? yield of extraction? purification process? This is currently missing from this section but is needed for full context before the characterisation of the compounds is presented. It is not enough to just present the methods in the experimental section. The authors should also expand the conclusions to expand on the potential impact of this work. They could also discuss some challenges and opportunities of drug discovery based on natural products

Author Response

Thanks for your kind suggestion, in our earlier version of the results and discussion part, it did seem kind of abrupt to jump right into the structural elucidation of compounds, thus in this version, we add paragraphs as follows:

“Based on our previous results from the same plant collected from Guizhou [10], flavonoids might be promising α-glucosidase inhibitors, and they are mainly present in petroleum ether and ethyl acetate parts extracted from 95% ethanol crude extract, thus herein the dried leaves of C. cajan (30 kg) were extracted with 95% ethanol. The extrac-tion solvent was then evaporated with a rotatory evaporator to give a crude extract, the crude extract was dissolved in water and successively partitioned with petroleum ether and ethyl acetate to afford petroleum ether part (835.6 g) and ethyl acetate part (456.4 g). Petroleum ether and ethyl acetate extracts being subjected to purification processes yielded ten stilbenoids (1-10) and seventeen flavonoids (11-27). The structures of new compounds (1-2) were characterized by HR-MS,1D-NMR, 2D-NMR, and ECD, while the structures of known compounds (3-27) were determined mainly through compari-son of 1D-NMR data with those reported in literatures.

In order to obtain promising α-glucosidase inhibitors, all the purified compounds were screened for their α-glucosidase inhibitory activity and the ones which showed robust activity were further elucidated for their inhibition type by Lineweaver-Burk plots.”

And in the end of the conclusion, we add sentences about the potential impact of expansion of the work, since it’s already lengthy, we didn’t go further on the challenges and opportunities drug discovery based on natural products, next time when preparing other articles, we will consider this. Thanks again for your advice on improving our manuscript. The paragraph we add at the end of the article is as follows:

“This study enriched the number of compounds isolated from the leaves of Cajanus cajan (L.) Millsp. and revealed some promising α-glucosidase inhibitors, which could be helpful for better utilization of this plant, yet more research is needed to demonstrate if the α-glucosidase inhibitors obtained herein could work in cell or animal model to treat diabetes.”

Reviewer 2 Report

In this article, the authors isolated two new compounds and several known stilbenoids and flavonoids from the extract of C. cajan leaves. The planar structures of the new compounds were elucidated and their absolute configurations were determined, and all the compounds were tested for α-glucosidase inhibition. Several comments are offered for authors’ consideration.

1. The experiment is complete, but the study is too simple, so it can be designed more deeply to make the study more scientific value.

2. The quality of the figures is low, such as Figure 5. Please increase the resolution.

3. More than half of the literature is out of date. Please avoid the use of too dated citations (e.g. the one dated 1974) in favor of more recent literature data available or justify their use.

4. Authors should correct carefully the formatting errors throughout this manuscript. e.g. the legend of figure 5.

Author Response

  1. The experiment is complete, but the study is too simple, so it can be designed more deeply to make the study more scientific value.

Response:

Thanks for your suggestion, we tried our best based on our limited available conditions, so right now that’s the best we could do, but we are sure our conditions will improve so will the quality of our work.

  1. The quality of the figures is low, such as Figure 5. Please increase the resolution.

Response:

We took your advice and re-saved our figures into clearer format, and rearrange their layout to make them seem more appropriate. We also make our figure 3 and figure 4 more clear and combine them into figure 3 to make it more good-looking.

  1. More than half of the literature is out of date. Please avoid the use of too dated citations (e.g. the one dated 1974) in favor of more recent literature data available or justify their use.

Response:

We changed too old literatures of year 1974, 1996, 2001, and 2004, others might not be new, but we used them only for data comparison and not for current research progress, they happened to have the data we wanted, so we didn’t change them, but we tried not adopting too old literatures since the data acquisition might be too different from ours due to the update of instruments for data collection.

  1. Authors should correct carefully the formatting errors throughout this manuscript. e.g. the legend of figure 5.

Thanks for pointing out our carelessness, we corrected this when rearrange the layout of figure 5 (new figure 4), and we also checked other formatting errors such as changing “trans, cis” into itallics and serial number of compounds into their bold form.

Reviewer 3 Report

The manuscript entitled Stilbenoids and Flavonoids from Cajanus cajan (L.) Millsp. And Their α-Glucosidase Inhibitory Activities” by Zhang et al. reported the isolation and structural elucidation of 2 new stilbene derivatives together with twenty-five  known compounds (327) from leaves of Cajanus cajan. Structures of new compounds were elucidated using MS, NMR and CD spectral data. Several compounds showed strong inhibitory activities against α-glucosidase, with compound 11 exhibiting the strongest activity. Furthermore, the enzyme kinetics of active compounds was studied.

Congratulation that the authors had very good results. The manuscript was prepared carefully. I recommend the manuscript can be accepted after minor revisions.

1.      Line 76: Rephrase this sentence “δC 168.4 signified a carbonyl group”

2.      Line 88-93,117-119: Correct “trans”, “cis”, “R”, “S” in italic 

3.      It’s better to move table 2 before part 2.2

4.      Line 302. Correct this sentence “experimental protocol is described in detail in 3.4”. Please add the reference and remove “in 3.4”.

5.      Please provide the 1H-1H COSY of compound 1 and 2.

6.      Why did the authors select leaves of Cajanus cajan for inhibitory activities against α-glucosidase ? Did they evaluate the α--glucosidase inhibitory activities of EtOH extract.

Author Response

  1. Line 76: Rephrase this sentence “δC4 signified a carbonyl group”

Response:

We rephrased the sentence to “The signal at δC 168.4 indicated the presence of a carbonyl group.”.

  1. Line 88-93,117-119: Correct “trans”, “cis”, “R”, “S” in italic 

Response:

We carefully read the whole manuscript and changed all the “trans”, “cis”, “R”, “S” into their italic form.

  1. It’s better to move table 2 before part 2.2.

Response:

We considered and agreed that it’s better to move table 2 before part 2.2, and renamed it as table 1.

  1. Line 302.Correct this sentence “experimental protocol is described in detail in 3.4”. Please add the reference and remove “in 3.4”.

Response:

We deleted in 3.4 and cite the literature 10.

  1. Please provide the 1H-1H COSY of compound 1 and 2.

Response:

We drew the 1H-1H COSY as red color bold line in figure 2 and their spectra are given in supporting information.

  1. Why did the authors select leaves of Cajanus cajan for inhibitory activities against α-glucosidase ? Did they evaluate the α--glucosidase inhibitory activities of EtOH extract.

Response:

We didn’t evaluate the α--glucosidase inhibitory activities of the EtOH extract, but we isolated the same plant from another place whose compounds have the activity, thus we assumed the compounds we isolated herein might also have the activity, we explained a little about this in this version in the results and discussion.

Other changes we think that might improve our manuscript are also made in this version, all the changes can be tracked in MS word “track changes mode”.

Round 2

Reviewer 2 Report

This article has not been substantially modified and is not recommended for publication.